# OpenReview forum: "Dataset and Lessons Learned from the 2024 SaTML LLM Capture-the-Flag Competition"
_NeurIPS.cc/2024/Datasets_and_Benchmarks_Track — NeurIPS 2024 Track Datasets and Benchmarks Spotlight_

### Official Review · Reviewer_uyPX · 2024-07-18
**Review for Paper #599**

**Rating:** 8
**Confidence:** 4
**Correctness:** Yes!
**Clarity:** Yes!

**Review:**

The paper is well-written but one of the major problems with the submission is that most of the content in the first eight pages has been attributed to the discussion of the defense and attack strategies used by the top three teams and little information has been shared about dataset insights. For instance, why do current models fail? Why do we observe different performances between gpt-3.5-turbo-1106 and llama-2-70b-chat, etc? In addition, please see the below points for improvement:

1. The competition used GPT-3.5 and Llama-2 models, which, while significant, may not fully represent the diversity of LLMs in use today. Findings may not generalize to other models with different architectures or training datasets. Since the authors introduce this as a benchmark it would be great if they could provide insights on other state-of-the-art LLMs.

2. Since every defense was bypassed at least once, it highlights the challenge and suggests that the defenses were perhaps not as strong as needed, which might reflect on the initial design of the defenses rather than the inherent difficulty of the problem.

3. As LLMs evolve and new models are developed, will the relevance and applicability of the findings diminish without continuous updates?

**Strengths:**

1. Creating a dataset with over 137,000 multi-turn attack chats provides a valuable resource for future research. In addition, the dataset is one of the largest and most comprehensive datasets available for studying prompt injection attacks on LLMs.

2. By focusing on practical security challenges in LLMs, the competition addresses urgent issues faced by the AI community, especially with the increasing deployment of LLMs in various applications.

3. The open-sourced platform allows researchers to benchmark new defense mechanisms against a standard dataset, promoting consistency and comparability in future research efforts.

**Additional Feedback:**

N/A

**Documentation:**

Yes!

**Ethics:**

1. While the research aims to enhance security, the detailed focus on attack methodologies could be misused by malicious actors if not properly controlled and monitored.

2. Highlighting vulnerabilities extensively might lead to increased exploitation before robust defenses are widely adopted, posing a risk to systems currently in use.

**Limitations:**

Please see the review section for more details.

**Opportunities For Improvement:**

Please see the review section for more details.

**Relation To Prior Work:**

Yes!

**Summary And Contributions:**

The paper details a capture-the-flag competition event held at IEEE SaTML 2024, which focused on identifying security risks in large language models (LLM). The competition aimed to uncover vulnerabilities related to prompt injection attacks. It consisted of two phases: a defense phase, where teams designed measures to protect secret strings within system prompts, and an attack phase, where teams attempted to extract these secrets from other teams' defenses. The findings revealed that every defense was bypassed at least once, underscoring the challenges in creating robust security for LLMs. The paper also introduces a comprehensive dataset of over 137,000 attack chats and makes the competition platform open-source to aid future research in LLM security.

---

> ### Author Rebuttal · Authors · 2024-08-15
>
> We thank you for your feedback and for your questions on dataset insights and for your suggestions on how to improve our work. We address your concerns as follows:
>
> > The paper is well-written but one of the major problems with the submission is that most of the content in the first eight pages has been attributed to the discussion of the defense and attack strategies used by the top three teams and little information has been shared about dataset insights.
>
> We would like to clarify our submission's alignment with the Datasets and Benchmarks track. The call for papers includes competition reports as part of its scope:
>
> _"In-depth analyses of machine learning challenges and competitions (by organisers and/or participants) that yield important new insight."_
>
> Given this context, we tried our best so that our submission:
>
> 1. Provides valuable insights from the competition and helps readers better understand the vulnerabilities of LLMs against prompt injections.
> 2. Releases a dataset that can be used to analyze attack/defense dynamics, benchmark defenses against prompt extraction attacks and train future models.
> 3. Open-sources a platform to reproduce our competition in the future with newer models and defenses.
> 4. Shares insights for future competitions organizers in related domains.
>
> This being said, we would be happy to include more dataset insights into the report. Does the reviewer have any specific insights in mind that could be useful for the reader to better understand the findings?
>
> > Since every defense was bypassed at least once, it highlights the challenge and suggests that the defenses were perhaps not as strong as needed, which might reflect on the initial design of the defenses rather than the inherent difficulty of the problem.
>
> Defenses in real world scenarios face the challenge of being robust to any future adversary, who might employ attack techniques not known by the defender when the defense gets initially deployed. This same challenge holds true in our competition, since defenses had to be submitted before the attack phase began and could not be updated adaptively.
>
> The competition was open to anyone, and several researchers in the fields of both machine learning and security took part in it. They had around one month to come up with good defenses, in a very restricted and simple scenario where they had to protect a short, well-defined secret with known format. Defenders had time to stress-test their own defenses, and it is very likely that they could not come up with an attack effective against their own defense—not even the defenders who turned out to be the most successful attackers! This suggests that the problem is indeed inherently difficult, with the constraints of the defense only having API access to the model.
>
> Moreover, even if all defenses were bypassed, some defenses were only broken once, demonstrating that at least some defenses were quite robust in this setup. To conclude, this shows that the community requires datasets and competitions like this to learn how effective attacks work and to develop more robust defenses. This can also include adaptive defenses to improve the robustness but also to to learn more about potential attack strategies against diverse defenses.
>
> >  For instance, why do current models fail? Why do we observe different performances between gpt-3.5-turbo-1106 and llama-2-70b-chat, etc?
>
> We thank the reviewer for pointing out this observation. We would like to refer you please to the discussion in page 17 “More powerful models are not necessarily more robust.”. In short, gpt-3.5-turbo-1106 is able to perform more sophisticated manipulation with text (e.g. writing a space in between each letter of the output). Therefore, once the model have been compromised, it is also able to better encode the secret in various ways that allows the attacker to circumvent the python and LLM filters (e.g., ascii encoding, embed the characters of the secret in a different context, etc.) We will add a clearer reference to this discussion from the main text.
>
> > As LLMs evolve and new models are developed, will the relevance and applicability of the findings diminish without continuous updates?
>
> Yes, this is a possibility, and is a downside of this class of paper/dataset in general.
> This is not specific to our setup, but to the entire idea of understanding the attack/defense frontier via an adversarial competition. The attacks must be necessarily adaptive (otherwise we are overestimating defense capabilities); and adaptivity relies on methods that are specific to the attacked system. Thus, changes in the system (e.g. newer models or post-training algorithms) will probably change the dynamics. However, we argue that these competitions and datasets still help the community gain useful insights about the state-of-the-art of the adversarial robustness of the systems and which lines of mitigations are more likely to be easily circumvented to avoid having a false sense of security. We open-source all our resources and future iterations of this competition could collect insights over time for newer LLMs, attacks and defenses.

---

> > ### Comment · Reviewer_uyPX · 2024-08-19
> >
> > Thank you for providing a detailed rebuttal response. The authors clarified all my concerns and pointed me to the correct sources/answers as needed. I am happy to raise my score to "Accept". Congratulations on a well-organized competition and documenting it into this nice paper!

---

### Official Review · Reviewer_VFHN · 2024-07-24
**Nice dataset collected from SaTML LLM CTF competition and a summary of the solutions.**

**Rating:** 8
**Confidence:** 3
**Correctness:** Yes
**Clarity:** Yes

**Review:**

See limitations and strengths

**Strengths:**

- Building on the popularity of the competition, the benchmark is well-designed with diverse, high-quality data.
- The analysis of the top solutions offers a deeper understanding of prompt injection attacks, particularly in scenarios involving well-designed defenses.

**Additional Feedback:**

N/A

**Documentation:**

Yes

**Limitations:**

- The dataset released by this work is not commonly used, and the authors do not specify its intended usage. Is it solely for analyzing attacks and defenses? Considering the possibility of more powerful attacks or defenses emerging in the future, will the dataset be extended, or will there be a procedure for people to submit their data to the dataset?
- As mentioned in the paper, the benchmark could be improved by using decoys to proactively counteract attackers.

**Opportunities For Improvement:**

- Consider building a continually updating benchmark that allows users to add their own attack and defense solutions.
- Include a greater variety of models and analyze the influence of these different models.
- Personally, I am also interested in the details of utility evaluation, which are not included in the paper. Given the diversity of filters, it would be interesting to understand how to establish a fair and effective utility evaluation.

**Relation To Prior Work:**

Yes

**Summary And Contributions:**

This paper primarily discusses the 2024 SaTML LLM Capture-the-Flag competition and releases the entire dataset collected from the event. Additionally, the authors analyze the top solutions employed during the competition and highlight future directions in this field.

---

> ### Author Rebuttal · Authors · 2024-08-15
>
> We thank you for the positive assessment of our work and for your suggestions on how to improve our submission. We address your concerns as follows:
>
> > The dataset released by this work is not commonly used, and the authors do not specify its intended usage. Is it solely for analyzing attacks and defenses? Considering the possibility of more powerful attacks or defenses emerging in the future, will the dataset be extended, or will there be a procedure for people to submit their data to the dataset?
>
> The dataset was released shortly after submission and has received over 400 downloads in the last month. We hope this dataset is not only useful for analyzing attacks and defenses, but it can also be used as a benchmark for future prompt injection defenses and potentially used to increase the safeguards of models via finetuning, as discussed in Section 7 in the paper. For example, conversations in the dataset can be used to train a model using instruction hierarchy (https://arxiv.org/abs/2404.13208) which did not release a public dataset, or to complement RLHF training.
>
> The dataset can be easily extended with more defenses and we also provide the codebase to reproduce our competition for additional data collection: https://github.com/ethz-spylab/satml-llm-ctf.
>
> > As mentioned in the paper, the benchmark could be improved by using decoys to proactively counteract attackers.
>
> Our dataset already includes defenses with decoys since many teams used this approach. Section 7 outlines learned lessons (from what we have already observed from the submitted attacks and defenses) in addition to future extensions.

---

> > ### Comment · Reviewer_VFHN · 2024-08-31
> > **Responese**
> >
> > Thanks for the response from the authors! This is the first dataset research on prompt injection, I hope the platform and dataset can lead future research!

---

### Official Review · Reviewer_E9B1 · 2024-07-25
**Red & Blue teaming for LLMs**

**Rating:** 9
**Confidence:** 5
**Clarity:** Yes. Well written and clearly articul…

**Review:**

The scoring was very evenly balanced for both the Red and Blue teams, ensuring that either side has their fair success.
The decoy techniques implemented by winning defense team has been discussed in some publications in the past but this was a more advance implementation of not only putting in decoys, but also adding filters on top of it.
Winning red team's approach of adaptive attacks and multi-turn evaluation showed the extent till which LLMs can be circumvented to bypass the built defenses and reveal the secret.

**Strengths:**

focus on both attack and defense.
giving enough abilities to both teams to implement their strategies.
evaluating both teams fairly.
Ensuring datasets are well labeled for each transaction.
Open sourcing their platform where the competition was conducted.
Novel techniques used to host the competition.

**Additional Feedback:**

NA

**Correctness:**

yes. evaluation of teams, collection of datasets and their platform structure looked correct to me.

**Documentation:**

yes.

**Ethics:**

No I dont see any ethics issues.

**Opportunities For Improvement:**

more specifications on the LLMs used, or finding a way to also rotate between LLMs in the competition could be helpful but it will add to the complexity of the process.

**Relation To Prior Work:**

yes.

**Summary And Contributions:**

Unlike traditional CTFs where the challenges are designed by the hosts and the participants spend time capturing the flag, this competition involved both defense and attack strategies to be implemented by participating teams. This led to a more broader view on the possible defense scenarios and how generic attack prompts could be created to bypass them.

---

> ### Author Rebuttal · Authors · 2024-08-15
>
> Thank you for the valuable feedback and a positive evaluation of our work. We address your concerns as follows:
>
> > more specifications on the LLMs used, or finding a way to also rotate between LLMs in the competition could be helpful but it will add to the complexity of the process.
>
> We agree that some detailed specifications on the used LLMs are helpful for the dataset. We will add this information to the paper in the coming weeks of the rebuttal. Rotating LLMs during the evaluation might be a potential safeguarding mechanism, however, for an attacker it would be enough to find an attack that is successful against only one LLM and try their attack until the system rotates to the desired LLM. Moreover, we have not considered this in our competition due to this not being common practice (because it can decrease utility in benign tasks).

---

### Decision · Program_Chairs · 2024-09-26

**Decision:**

Accept (Spotlight)

**Comment:**

This paper presents a comprehensive study from a recent CTF competition at IEEE SaTML (Secure and Trustworthy Machine Learning) 2024. It makes the following contributions: (1) the paper studies both the attacks and defenses about exposing the secret string from the LLM system prompt. It summarizes some interesting insights and lessons from this competition. (2) the authors established and open-sourced a dataset with a large number of multi-turn attack chats, which can benefit the future studies. All the reviewers appreciated the significance of this study, findings and released dataset, and the majority of their concerns have been addressed by the authors' responses. Therefore, this paper is recommended for acceptance. I suggest the authors to keep maintaining/updating the dataset for a wider range of usage in the community.